# Immunoinformatics-Aided Design and Evaluation of a Potential Multi-Epitope Vaccine against *Klebsiella Pneumoniae*

**DOI:** 10.3390/vaccines7030088

**Published:** 2019-08-12

**Authors:** Hamza Arshad Dar, Tahreem Zaheer, Muhammad Shehroz, Nimat Ullah, Kanwal Naz, Syed Aun Muhammad, Tianyu Zhang, Amjad Ali

**Affiliations:** 1Atta-ur-Rahman School of Applied Biosciences (ASAB), National University of Sciences and Technology (NUST), Islamabad 44000, Pakistan; 2Institute of Molecular Biology and Biotechnology, Bahauddin Zakariya University, Multan 60800, Pakistan; 3State Key Laboratory of Respiratory Disease, Guangzhou Institutes of Biomedicine and Health (GIBH), Chinese Academy of Sciences (CAS), Guangzhou 510530, China

**Keywords:** *Klebsiella pneumoniae*, pangenome, reverse vaccinology, potential vaccine candidate, immunoinformatics, multi-epitope vaccine

## Abstract

*Klebsiella pneumoniae* is an opportunistic gram-negative bacterium that causes nosocomial infection in healthcare settings. Despite the high morbidity and mortality rate associated with these bacterial infections, no effective vaccine is available to counter the pathogen. In this study, the pangenome of a total of 222 available complete genomes of *K. pneumoniae* was explored to obtain the core proteome. A reverse vaccinology strategy was applied to the core proteins to identify four antigenic proteins. These proteins were then subjected to epitope mapping and prioritization steps to shortlist nine B-cell derived T-cell epitopes which were linked together using GPGPG linkers. An adjuvant (Cholera Toxin B) was also added at the N-terminal of the vaccine construct to improve its immunogenicity and a stabilized multi-epitope protein structure was obtained using molecular dynamics simulation. The designed vaccine exhibited sustainable and strong bonding interactions with Toll-like receptor 2 and Toll-like receptor 4. *In silico* reverse translation and codon optimization also confirmed its high expression in *E. coli* K12 strain. The computer-aided analyses performed in this study imply that the designed multi-epitope vaccine can elicit specific immune responses against *K. pneumoniae*. However, wet lab validation is necessary to further verify the effectiveness of this proposed vaccine candidate.

## 1. Introduction

*Klebsiella pneumoniae* (*K. pneumoniae*) causes nosocomial infections, majorly infecting immunocompromised patients. A number of pathological conditions are associated with this bacterial infection such as pneumonia, bacteremia, septicemia, endocarditis, meningitis, and cellulitis [1]. Especially, carbapenemase-producing *K. pneumoniae* strains present a major problem from a clinical and epidemiological perspective. Mortality rates associated with carbapenemase-producing *K. pneumoniae* infections reportedly vary from 22% to 72%, thus underlying the significant threat to human health posed by this opportunistic microorganism [2,3,4]. 

Lipopolysaccharide (LPS) of *K. pneumoniae* has been used previously to develop vaccine but severe side effects associated with LPS-containing vaccines present a major problem [5]. Capsular polysaccharides (CPS) have also been explored for vaccine preparation but CPS-based vaccines, although nontoxic and immunogenic, contain many K-types (77 different antigens), which proves a daunting challenge [6]. Thus, a broad-spectrum vaccine needs to be urgently designed and developed against *K. pneumoniae*.

Immunization studies mostly involve the culturing of microorganisms, isolation of pathogen-specific antigen, and inactivation and reinjection into subjects to confirm immune response [7]. The procedure is cost ineffective and requires a lot of time. Moreover, these traditional methods are not appropriate to work against *K. pneumoniae*, which includes a variety of strains. Undoubtedly, the associated protocol would require the exhaustive search of finding a conserved antigen, processing it, and assessing it for vaccine efficacy. Advancements in the field of DNA sequencing, systems biology, genomics, and proteomics have enabled a greater understanding of pathogenesis and vaccine design [8]. All antigens of a pathogen can be tested easily to evaluate their immunogenicity. Reverse vaccinology is a method that performs genome mining using computer-aided analyses for the identification of potential vaccine candidates (PVCs) [9]. In this study, we have integrated pangenomics, reverse vaccinology, and immunoinformatics approaches for the design and evaluation of a multi-epitope vaccine against *K. pneumoniae*. A pangenome approach has been applied for the identification of core proteins that can be used for developing broad spectrum therapeutics against bacterial infections [10]. It is important that the prioritized therapeutic targets are vital for the survival of pathogen and they are not human-homologs [11]. Furthermore, extracellular and secretory proteins are likely to be vaccine candidates [12].

Ideally, a vaccine should contain conserved epitopes that in turn can generate specific B-cell and T-cell (CD4 and CD8) responses [13,14,15]. This will enable the induction of pathogen-specific immune responses with minimal side effects by incorporating highly conserved epitopes in a vaccine formulation [16]. Using immunoinformatics approaches, the structure of the multi-epitope vaccine was modeled, and molecular refinements were applied. Finally, the vaccine was docked with TLR2 and TLR4 to decipher various interactions between the designed vaccine and immune receptors.

## 2. Materials and Methods 

### 2.1. Genome and Proteome Retrieval and Pangenome Analysis

A total of 222 complete genome and proteome sequences of *K. pneumoniae* present in the NCBI (National Center for Biotechnology Information) genome list were acquired on 4th September 2018. Pangenome analysis on all the strains was conducted using the BPGA (Bacterial Pan Genome Analysis) tool using USEARCH clustering algorithm [17,18]. For clustering orthologous proteins, a 90% sequence identity threshold was selected due to the focus on highly conserved proteins present in the 222 strains of *K. pneumoniae*.

### 2.2. Reverse Vaccinology for Protein Prioritization

Protein prioritization was conducted using a reverse vaccinology strategy on the core proteins. For this purpose, VacSol in-house pipeline was used [19]. Firstly, non-human homologs were selected to minimize the chances of autoimmunity or low immune response/tolerance [20]. The exclusion criteria chosen was a percentage identity greater than 35%, Bit Score > 100 and E-Value < 1.0 E^−5^. Next, the non-human homologs were subjected to subcellular localization using Psortb version 2.0 to positively select extracellular and outer membrane proteins as they are accessible to the host immune system [21,22]. Selected proteins were subjected to BLASTp against the DEG (Database of Essential Genes) for the identification of essential proteins as they perform major cellular functions of bacteria [23,24]. Next, virulence factors involved in pathogenesis and disease development were retained using BLASTp against the Virulence Factor Database (VFDB) and Microbial virulence Database (MvirDB) [25,26]. In addition, proteins were checked for the number of transmembrane helices using HMMTOP version 2.0, and those containing <2 helices were selected to ensure proper purification of putative proteins in wet lab studies [27,28]. Finally, low molecular weight proteins were shortlisted using the ProtParam tool in the ExPASy portal to ensure easy protein purification in vaccine development [29,30,31].

### 2.3. Prediction of B-Cell Epitopes

ABCPred was used to identify potential B-cell epitopes in prioritized proteins [32]. The server uses an artificial neural network to provide information about putative B-cell epitopes. Default parameters were applied and B-cell epitopes above 0.51 threshold were subjected to T-cell epitope prediction.

### 2.4. Prioritization of B-Cell Derived MHC II Epitopes

ProPred was used to search CD4+ T-cell epitopes within selected B-cell epitopes using default settings [33]. The server allows the evaluation of T-cell epitope binding to 51 different MHC II alleles. MHC II epitopes projected to associate with at least ten different MHC II alleles were retained. Antigenicity potential of the selected MHC II epitopes was determined using VaxiJen v2.0, which uses an alignment-free approach to determine probable protective antigens [34]. MHC II epitopes possessing scores more than 0.5 were retained as antigenic epitopes. Further, the epitopes were evaluated for their binding affinity with predominant HLA II allele DRB1*0101 using MHCPred server version 2.0 [35]. For this, epitopes with IC50 value < 500 nM were shortlisted as this is an established threshold associated with immunogenicity [36]. MHC II epitopes were scrutinized based on their virulence potential using VirulentPred [37]. Finally, MHC II epitopes were evaluated for their ability to induce IFN-γ using an IFNepitope server [38]. Only those epitopes predicted to induce IFN-gamma were prioritized accordingly.

### 2.5. Prioritization of B-Cell Derived MHC I Epitopes

ProPred I was used to search for potential MHC I epitopes within B-cell epitopes [39]. The option of immunoproteasome and proteasome filters was selected to maximize the chances of finding CD8 epitopes. ProPred I allows the prediction of epitope binding to 47 different MHC I alleles. Only epitopes found to associate with at least five different MHC I alleles were considered for further analysis. Predicted MHC I epitopes were assessed for their immunogenic ability using the MHC I immunogenicity score provided by the IEDB (Immune Epitope Database) web resource [40]. The MHC I immunogenicity tool in the IEDB exploited the properties and positions of amino acids within the epitopes to predict MHC I immunogenicity. The tool has been validated for 9-mer peptides. Further, the epitopes were evaluated for their binding affinity with predominant HLA I allele HLA A*0201 using MHCPred server version 2.0 [35]. For this, epitopes with IC50 value < 500 nM were shortlisted as this is an established threshold associated with immunogenicity [36]. Finally, MHC I epitopes were prioritized based on their virulence potential using VirulentPred [37]. 

### 2.6. Multi-Epitope Vaccine Design

The prioritized epitopes were linked together and with an immunological adjuvant to construct a multi-epitope vaccine. To enhance the immunogenicity of the vaccine, the amino acid sequence of cholera toxin subunit B (CTB) was attached to the N-terminal end of the vaccine construct using an EAAAK linker. CTB is the nontoxic portion of cholera toxin and shows specific affinity to the monosialotetrahexosylganglioside (GM1) on gut epithelial cells and various other cell types such as antigen presenting cells, i.e., dendritic cells, macrophages, and B cells, which allows its maximum exposure and accessibility to the immune system [41]. A flexible glycine-proline linker GPGPG was used to join adjacent epitopes for efficient separation and to avoid the possibility of junctional epitope formation. 

### 2.7. Antigenicity, Allergenicity, Solubility, and Physicochemical Features

The finalized sequence of the poly-epitope vaccine candidate was checked for antigenicity, allergenicity, solubility, stability, molecular weight etc. Antigenicity of the vaccine was calculated using ANTIGENpro and VaxiJen 2.0 [34,42]. The accuracy of ANTIGENpro on the combined dataset was found to be 76% using cross-validation experiments; meanwhile, the VaxiJen server’s performance accuracy varied from 70% to 89%, depending on the target organism selected. AllergenFP version 1.0 and AllerTOP version 2.0 were used to determine vaccine as allergen or non-allergen [43,44]. SOLpro calculated the propensity of protein to be soluble in *E.coli* on overexpression [45]. Additionally, PROSO II was also used to cross-check SOLpro results [46]. A FASTA sequence of vaccine candidate was then used to calculate isoelectric point, molecular weight, and other parameters like half-life, instability index, aliphatic index, GRAVY, and so on from the ExPASy server [29]. TMHMM version 2.0 was used to search potential transmembrane helices in the final vaccine construct [47]. Additionally, SignalP 4.1 was used to find any potential signal peptides in the vaccine [48].

### 2.8. Multi-Epitope Structural Modeling, Refinement, and Validation

The putative 3D structure of vaccine protein was generated using 3Dpro [45]. Molecular refinements were applied to the modeled structure using the GalaxyRefine server [49]. The server subjected the modeled vaccine to structural perturbations followed by structural relaxations. For model 1, structure perturbation is applied exclusively to side-chain clusters, and for model 2‒5, more aggressive perturbations are applied. Each of the five models were checked for GDT-HA, RMSD, MolProbity score, etc, and the best model was selected to proceed. The shortlisted model was evaluated using Ramachandran plot analyses, ERRAT score, and Prosa-generated Z-scores for confirmation [50,51,52].

### 2.9. Energy Minimization of Multi-Epitope Vaccine

A molecular dynamics simulation technique was applied on the vaccine for energy minimization. For this, a command line Linux-based program, GROMACS (GROningen MAchine for Chemical Simulations), was used [53]. The vaccine structure was subjected to molecular simulations to imitate the biological environment faced by protein structure inside the biological system. 

Using pdb2gmx, a gro file was generated from the structure of the vaccine to obtain the topology compatible with the OPLS-AA (Optimized Potential for Liquid Simulation-All Atom) force field [54]. The structure was confined in a rhombic cube to fill water molecules. Then, it was positioned in the box’s center at 1 nm from the edge of the cube to generate its periodic image 2 nm apart. Water was used to simulate the vaccine, which was spatially placed with a force constant of 1000 kJ mol^−1^nm^−2^. The total charge on the vaccine was calculated. The system was neutralized by adding charge and an ions.mdp file was generated. The Verlet scheme was selected and electrostatic forces were enabled. The process of energy minimization was conducted, and the energy minimized structure was obtained. To stabilize the temperature, NVT equilibration was performed for 100 ps. Several simulations were conducted at different initial speeds, V-rescale was used, and the temperature fluctuations were observed from the temperature graph. NPT ensemble enabled the calculation of pressure and density; the process comprised 50000 steps. The resulting structure was subjected to molecular dynamics (MD) simulation for 10 ns. The RMSD of backbone of energy minimized structure was predicted and results were graphed. The radius of gyration (Rg) was calculated to get an insight into the compactness of structure. Simulation graphs were analyzed using the Xmgrace tool [55]. 

### 2.10. Binding Affinity of Poly-Epitope Structure with Toll-Like Receptors

The poly-epitope vaccine candidate was checked for binding affinity with TLR2 and TLR4 separately to validate its ability to be an agonist to both TLR2 and TLR4. In humans, *K. pneumoniae* infection leads to over-expression of TLR2 and TLR4 in airway epithelial cells [56]. TLR2 structure was obtained from PDB ID 2Z7X while that of TLR4 was extracted from PDB ID 3FXI. 

The structures of the vaccine and TLRs were submitted to HADDOCK server guru level interface using preselected parameters [57]. Guru level is an upgraded HADDOCK interface that permits the identification of flexible regions that are important for molecular docking. Docked clusters were formed and based on the lowest HADDOCK score, and one cluster was identified. A representative structure from this cluster was subjected to molecular refinements for better positioning. CPORT (consensus prediction of interface residues in transient complexes) predicted active and passive residues that are involved in interaction [58]. Chimera was used to visualize the refined TLR-vaccine structures [59]. To map the interacting residues between the vaccine and TLRs, PDBsum was used [60].

### 2.11. Reverse Translation and Codon Optimization

Reverse translation of the multi-epitope vaccine sequence was performed using online server JCat to obtain a cDNA sequence of the gene which was then subjected to codon optimization [61]. Codon optimization assessed the cDNA as per the GC content of the sequence and Codon Adaptation Index (CAI). GC content of the sequence should lie within the range of 30–70%; in contrast, CAI values vary from 0 to 1, with a greater CAI value indicating higher level of gene expression [62]. A CAI score of 1 is considered ideal, nevertheless, a score of more than 0.8 is still acceptable.

## 3. Results

### 3.1. Pangenome Analysis of Klebsiella Pneumoniae

A total of 222 complete genomes and associated proteomes of *K. pneumoniae* were downloaded from the NCBI. Their information such as accession number, strain name, and genome statistics are provided in Appendix A. The metadata associated with these bacterial strains such as isolation source and the country of isolation is provided in Appendix A. Pangenome analysis revealed that the core genome of *K. pneumoniae* consists of 2212 proteins, which were extracted for further processing.

### 3.2. Prioritization of Global Core Antigenic Proteins

A total of 1940 proteins from the core proteome were found to be non-human homologous proteins using BLASTp against human genome (Appendix A). Next, the non-human homologs were subjected to subcellular localization. As a result, 35 proteins were found to lie in the outer membrane and extracellular location (Appendix A). Proteins residing in these locations tend to participate in pathogenesis, thus they represent good targets for vaccine development [63]. The DEG predicted essential proteins after subcellular localization [23]. Results revealed that a total of 12 proteins were essential (Appendix A). Identification of virulence factors in bacteria is a key step in vaccine candidate identification [64]. Essential proteins were evaluated based on virulence properties and a total of four proteins were found to be associated with virulence (Appendix A). Next, a transmembrane topology filter was applied. VacSol detected all four proteins with less than 2 transmembrane helices. Presence of multiple transmembrane helices in a protein makes expression difficult and hinders purification of recombinant proteins for vaccine production [22]. 

Thus, VacSol prioritized a total of four proteins based on the non-homology to human, <2 transmembrane helices, essentiality, virulence, and subcellular localization predictions (Appendix A). According to UniProt annotation, the four putative targets are: outer membrane protein A, copper/silver efflux RND transporter outer membrane protein, phosphoporin PhoE, and peptidoglycan-associated lipoprotein (Pal). All the prioritized proteins were projected to lie in the outer membrane region and predicted to be antigenic by both AntigenPro and Vaxijen, thus strongly indicating their antigenic potential and their utility in future vaccine development against *K. pneumoniae*. The putative antigens are part of the core proteome and as such are highly conserved proteins that could be exploited for the development of broad-spectrum therapeutics, i.e., vaccines and/or drugs. Furthermore, they are virulent and essential for the survival of the bacteria. All the prioritized proteins have <110 kD molecular weight and thus ensure easy purification for vaccine development.

### 3.3. Selection of Epitopes from Global Core Antigenic Proteins

A total of 135 linear B-cell epitopes were predicted by using ABCPred server default parameters (Appendix A sheet 1). T-cell epitopes were searched within B-cell epitopes to define B-cell derived T-cell epitopes. A total of 40 B-cell derived MHC II epitopes were identified to associate with ≥10 MHC II alleles (Appendix A sheet 2). Out of these, 20 antigenic epitopes were identified using VaxiJen (Appendix A sheet 3). Scrutinization based on an IC50 value <500 nM, i.e., good binding affinity with HLA II allele DRB1*0101, using MHCPred server version 2.0 removed one epitope from further consideration (Appendix A sheet 4). Finally, a total of five IFN-γ inducing MHC II epitopes were shortlisted (Appendix A sheet 5). In contrast, a total of 15 B-cell derived MHC I epitopes were identified that associated with ≥5 different MHC I alleles (Appendix A sheet 6). Out of these, a total of seven epitopes were found to have a positive MHC I immunogenicity score using the IEDB (Appendix A sheet 7). Finally, four MHC I epitopes were finalized based on an IC50 value <500 nM, i.e., good binding affinity with HLA A*0201, using MHCPred server version 2.0 (Appendix A sheet 8). All the prioritized B-cell derived T-cell epitopes were found to be virulent using VirulentPred.

### 3.4. Multi-Epitope Vaccine Design

The prioritized B-cell derived T-cell epitopes were joined together using GPGPG flexible linkers. The amino acid sequence of CTB adjuvant was added at the N-terminal of the vaccine construct by EAAAK linker. After adding adjuvant and linkers, the multi-epitope vaccine was found to be 230 amino acids in length. The sequence of the multi-epitope vaccine is provided: MTPQNITDLCAEYHNTQIHTLNDKIFSYTESLAGKREMAIITFKNGATFQVEVPGSQHIDSQKKAIERMKDTLRIAYLTEAKVEKLCVWNNKTPHAIAAISMANEAAAKLEYQWVNNIGPGPGLGVSYRFGQGPGPGYVRFGIKGEGPGPGGSFDYGRNLGPGPGNEITLFTALGPGPGDILEAEHSLGPGPGLQYQGKNEGGPGPGLVTATAGYQGPGPGGLMIALPVM.

### 3.5. Physicochemical and Other Evaluations of the Multi-Epitope Vaccine

A physicochemical feature check revealed that the designed vaccine has a molecular weight of 24.221 kD. Proteins having <110 kD molecular weight are believed to be good vaccine candidates [22]. Estimated half-life was found to be greater than 7.2 hours (mammalian reticulocytes, in vitro), greater than 20 hours (yeast, in vivo), and greater than 10 hours (*Escherichia coli*, in vivo). The instability index was calculated to be 27.69 (<40), hence the vaccine is considered stable. The aliphatic index was found to be high, i.e., 73.87, hence the vaccine is projected to be thermostable at various temperatures. The grand average of hydropathicity (GRAVY) value was computed and found to be -0.318, hence the vaccine is hydrophilic and thus likely to interact with water molecules. Therefore, this analysis indicates that the designed multi-epitope vaccine is physiochemically appropriate for vaccine production. Both VaxiJen and AntigenPro predicted the antigenic nature of the vaccine construct. Computational analysis projected non-allergenicity, thus the vaccine is not expected to trigger harmful allergic responses in humans. According to the SOLpro and PROSO II results, the vaccine is soluble. No signal peptide was detected in the designed construct and no transmembrane helix was projected so no expression difficulties are anticipated in the development of the vaccine.

### 3.6. Modeling and Refinements of the 3D Structure of the Vaccine

The three-dimensional structure of the vaccine was constructed using a 3Dpro ab initio predictor [45]. 3Dpro was suitable to model the vaccine structure due to the lack of good structural templates for homology modeling [45]. The predicted model was then subjected to structure perturbations and relaxations to attain a refined structure using GalaxyRefine (Figure 1A) [49]. RAMPAGE server was used to determine the quality of the refined tertiary structure [51]. Ramachandran plot analysis by this server showed that 94.7% of the residues of the vaccine are present in the favored region while 2.6% of the residues are present within the allowed and outlier regions; further, the ProSA Z-score was calculated to be ‒1.86, which is in the vicinity of experimental structures (Figure 1B,C) [52]. ERRAT projected an overall quality score of 78.2609, which further supported the high-quality structure of the refined model [50].

### 3.7. Molecular Dynamics Simulation of the Multi-Epitope Vaccine

The molecular dynamics (MD) simulation technique was applied to the three-dimensional structure obtained in the previous step. An OPLS-AA force field was applied and the mass of the vaccine construct after applying the force field was found to be 24219.539 amu. A total of 32473 water molecules were added into the system using the built-in spectro tool of GROMACS (GROningen MAchine for Chemical Simulations). The total charge on protein was calculated to be ‒2. To neutralize the system, two positive sodium ions were added that replaced two existing water molecules at atoms 13165 and 30499. After this addition of positive ions, a total of 32471 water molecules remained.

Energy minimization was performed for 50000 steps; upon reaching 1657 steps, the steepest descents converged, and the force reached <1000 KJ/mol. The potential energy was found to be −1.7771606e+06 kJ/mol, while the average potential energy of the system during energy minimization was found to be −1.72744e+06 kJ/mol with a total drift of −173183 kJ/mol. The average temperature after 50000 steps was 299.77 K (Figure 2A); the average pressure after 50000 steps of NPT was −2.44361 bar with the total drift of −12.0214 bar (Figure 2B), while the average density was found to be 1008.82 kg/m^3^ with a total drift of 0.788372 kg/m^3^.

Trajectory analysis was conducted after 10 ns simulation time. The radius of gyration reached about 2 nm, signifying that the three-dimensional protein structure remained stable during MD simulation (Figure 2C). A plot of RMSD backbone revealed that RMSD levels go up to ~0.6 nm and are maintained during most of the simulation time, signifying that the vaccine remains stable over time (Figure 2D). RMSF, on the other hand, explains regions with high flexibility (Figure 2E). 

### 3.8. Molecular Docking of Vaccine with Toll-Like Receptors

The HADDOCK (High Ambiguity Driven protein-protein DOCKing) server performed docking of the vaccine construct with TLR2 and TLR4. In the case of TLR2 docking, HADDOCK grouped 99 structures into 13 clusters, which represents 49.5% of the water-refined models HADDOCK generated. The top cluster having the lowest HADDOCK score, i.e., −23.6 +/− 21.5, is the most reliable amongst all clusters. A representative model from this top cluster was subjected to refinement accordingly. The HADDOCK refinement server clustered the resulting 20 structures into a single cluster, which represents 100% of the water-refined models HADDOCK generated. The statistics from this refined cluster are shown in Table 1, and details of individual parameters are provided in Appendix A.

In the case of TLR4 docking, HADDOCK grouped 102 structures into 14 clusters, which represented 49.5% of the water-refined models HADDOCK generated. The top cluster having the lowest HADDOCK score, i.e., −4.8 +/− 48.2, is the most reliable amongst all clusters. A representative model from this top cluster was subjected to refinement accordingly. The HADDOCK refinement server grouped the resulting 20 structures into a single cluster, which represents 100% of the water-refined models HADDOCK generated. The statistics from this refined cluster are shown in Table 2 and details of individual parameters are provided in Appendix A.

The analysis of docking results revealed good interaction between the vaccine construct and TLR2/TLR4 (Figure 3A and Figure 4A). To get detailed insights into interface residues and intermolecular binding forces between the docked complex, the PDBsum online server was used. A total of 45 interface residues of the multi-epitope vaccine associated with a total of 37 residues of TLR2 (Figure 3B). The interface area (Å2) of the vaccine was found to be 2109, while that of TLR2 was found to be 2176. Likewise, collectively 36 interface residues of the multi-epitope vaccine associated with a total of 38 residues of TLR4 (Figure 4B). The interface area (Å2) of the vaccine was found to be 1983, while that of TLR4 was found to be 1950. 

Our vaccine formed a total of 17 hydrogen bonds [Chain A(Vaccine)-B(TLR2); 9-109, 10-85, 31-32, 35-31, 35-37, 37-38, 44-29, 44-39, 64-30, 64-31, 71-39, 90-111, 182-396, 183-422, 184-422, 185-445, and 186-424] with TLR2. Structural analysis showed that Leu9 and Tyr109 formed a hydrogen bond at a distance of 2.78 Å; similarly, Cys10-Ser85 developed a hydrogen bond at 2.73 Å. Ser31-Arg32 forms a hydrogen bond at the distance of 2.75 Å. Likewise, Lys35-Asp31 forms a bond at 3.07 Å, Lys35-Lys37 at 2.69 Å, Glu37-Gly38 at 3.26 Å, Lys44-Ser29 at 2.88 Å, Lys44-Ser39 at 2.66 Å, Lys64-Cys30 at 2.93 Å, Lys64-Asp31 at 2.73 Å, Asp71-Ser39 at 2.67 Å, Asn90-Tyr111 at 3.13 Å, Leu182-Gln396 at 3.01 Å, Glu183-Lys422 at 2.95 Å, Ala184-Lys422 at 3.12 Å, Glu185-Ser445 at 3.07 Å, and His186-Ser424 at 2.82 Å.

The vaccine formed a total of 13 hydrogen bonds [Chain A(Vaccine)-B(TLR4); 7-547, 12-477, 32-606, 35-550, 44-601, 44-602, 63-608, 64-603, 64-605, 85-596, 89-569, 92-569 and 185-153]. Structural analysis showed that Thr7 develops a hydrogen bond with Gln547 at a distance of 2.9 Å, whereas Glu12 forms a hydrogen bond with Lys477 at 2.6 Å. Similarly, Leu32-Arg606 forms a hydrogen bond at a distance of 2.9 Å, Lys35-Asp550 at 2.6 Å, Lys44-Leu601 at 3.1 Å, Lys44-Val602 at 2.8 Å, Lys63-Glu608 at 2.8 Å, Lys64-Glu603 at 2.6 Å, Lys64-Glu605 at 2.6 Å, Lys85-Asp596 at 2.7 Å, Trp89-Ser569 at 3.2 Å, Lys92-Ser569 at 2.7 Å, and Glu185-Lys153 at 2.7 Å.

Upon close observation, it was revealed that most of the distances of the hydrogen bonds between the vaccine and TLR2 and TLR4 lie within the range of 2‒3 Å, thus indicating strong interactions [65]. Therefore, it is expected that our designed vaccine construct will associate with both TLR2 and TLR4, leading to the induction of associated immune pathways. 

### 3.9. Reverse Translation and Codon Optimization of Vaccine

Reverse translation and codon optimization were performed on the sequence of the multi-epitope vaccine using the JCat server [61]. In this way, the CAI and GC content were determined. The GC content was found to be 54.49%, which is appropriate as it lies within the range 30‒70%. The CAI value was calculated to be 1, which projects high level expression of our designed vaccine construct in *E. coli* K12 strain.

## 4. Discussion

A large-scale comparative genome analysis of *K. pneumoniae* is required to understand its global epidemiology, antimicrobial resistance, virulence, and phylogeny. The reverse vaccinology strategy adopted in this study will be helpful in the development of specific therapies against various *K. pneumoniae* strains. Despite some progress in the development of therapies to counter *K. pneumoniae* infections until now, a safe and effective vaccine is highly desirable to prevent the spread of this opportunistic pathogen. Two bacterial surface antigens, LPS and CPS, have been mainly exploited for this purpose [5,66]. However, severe side effects associated with LPS-based vaccines render them unfavorable for vaccine development. In contrast, CPS-based vaccines are both nontoxic and immunogenic; nonetheless, the increased number of different capsular K-type antigens (at least 77) makes it difficult to cover maximum bacterial strains [6,67]. A 24-valent CPS vaccine was developed and later shown to be safe and immunogenic [6]. However, the maximum protection coverage provided by this vaccine was not achieved in more than 70% of the bacterial strains despite many efforts [68].

The reverse vaccinology strategy adopted in this study can provide an effective therapeutic solution to the problem of genetic diversity both within *K. pneumoniae* and human species. This was done by considering only conserved bacterial proteins for target prioritization and by focusing on T-cell epitopes that can interact with a maximum number of HLA alleles. 

A total of four outer membrane proteins were shortlisted for vaccine designing. Outer membrane proteins (OMPs) are crucial for the transport of molecules, maintenance of membrane integrity, and pathogenesis [69]. Furthermore, outer membrane or secreted proteins are more likely to be good vaccine candidates as they are readily accessible to the host immune system [22].

Two of the prioritized antigens in this study, peptidoglycan-associated lipoprotein (Pal) and outer membrane protein A (OmpA), are recognized as dominant OMPs released by gram-negative bacteria during sepsis [70]. Kurupati et al. identified OmpA and other proteins using immunoproteome analysis to identify potential antigens of *K. pneumoniae* [71]. In another study, Kurupati et al. vaccinated mice using plasmid DNA containing *ompA* or *ompK36* of *K. pneumoniae*, which generated humoral as well as Th1 cell-mediated immune responses [72]. Similarly, Pal protein of *K. pneumoniae* is involved in protection against serum killing and neutrophil-mediated phagocytosis and plays an important role in preserving the integrity of cell envelope and imperative for *K. pneumoniae* virulence in vivo [73]. This study suggests that *K. pneumoniae* Pal can be used as an attenuated vaccine in the future.

The outer membrane porin-encoding gene, PhoE, is known as a housekeeping gene used for the molecular typing of *K. pneumoniae* [74]. According to information in the UniProt, PhoE expression is induced by phosphate starvation and is involved in the uptake of inorganic phosphate, phosphorylated compounds, and other negatively charged solutes. Considering these crucial functions, PhoE protein also represents a good therapeutic target to tackle *K. pneumoniae* infection. The fourth proposed candidate, CusC, is a part of the cation efflux system that exports silver or copper ions from the periplasm via an antiport mechanism [75]. This overcomes the challenge posed by therapeutic usage of silver and copper and may help *K. pneumoniae* survive under these stress conditions.

In this study, we used an in-house pipeline “VacSol” to identify potential antigens in *K. pneumoniae*, and the candidates were verified through existing experimental data where applicable. Based on our experience and existing knowledge, we believe that these predicted proteins are good vaccine candidates. They are highly conserved proteins present in the majority of, if not all, *K. pneumoniae* strains. Furthermore, they are localized in the outer membrane and thus accessible to the host immune system, are known to be associated with virulence, and are crucial for the survival of bacteria. A similar approach based on reverse vaccinology has also been applied for the identification of vaccine targets in *K. oxytoca*, another closely related species to *K. pneumoniae* [76]. Furthermore, this approach has been adopted to target other gram-negative pathogens such as *Acinetobacter baumannii* [28,77], *Pseudomonas aeruginosa* [78], *Neisseria gonorrhoeae* [79], etc.

Selected T-cell epitopes were found to be conserved among bacterial strains and predicted to be immunogenic in nature, thus they are perfect candidates for developing a broad-spectrum vaccine. IFN-γ inducing MHC II epitopes were also predicted in this study. The production of IFN-γ contributes to the clearance of *K. pneumoniae* infections, especially for hypervirulent strains [80,81]. IFN-γ is essential for the innate response against pulmonary *K. pneumoniae* infection [82]. Thus, the prioritized epitopes have the potential to generate more specific, effective, safe, and durable immune responses as well as avoid all the undesired effects.

Multi-epitope vaccines can activate both humoral and cellular immune responses and thus are considered a better alternative to monovalent vaccines [83]. In this study, only the pathogenic bacterial strains having complete genome sequences available were subjected to target prioritization and epitope-mapping to ultimately construct a multi-epitope vaccine comprising prioritized epitopes. Adjuvant was also added in the multi-epitope vaccine construct to enhance its immunogenicity. The physiochemical analyses of the vaccine projected it to be stable, hydrophilic, and acidic in nature. Structural evaluations using a Ramachandran plot and ProSA indicated that a high-quality model was obtained with near-native structure. 

Molecular docking was carried out to analyze the interaction of designed vaccine construct with immune receptors TLR2 and TLR4. *K. pneumoniae* infection leads to the overexpression of TLRs in human airway epithelial cells [56]. Further, the level of TLR activation determined the extent of TLR2 and TLR4 upregulation. Thus, TLR2 and TLR4 in humans mediate innate immune responses against invading *K. pneumoniae* pathogen. Hence, the designed vaccine was assessed for association with TLR2 and TLR4. The docking score revealed significantly high interaction between vaccine and innate immune receptors thus suggesting that the vaccine can trigger TLR activation and consequently lead to enhanced immune responses against *K. pneumoniae*. 

## 5. Conclusions

In this study, a reverse vaccinology approach was followed on 222 complete *K. pneumoniae* genomes to identify a total of four conserved antigenic proteins. Furthermore, a novel multi-epitope vaccine candidate containing high-ranked epitopes from all four antigens was constructed using immunoinformatics approaches and evaluated using MD simulations and protein-protein docking. The application of computational tools in vaccine design can significantly enhance the process of vaccine discovery and accomplish this goal in less time and with fewer costs. The designed vaccine has suitable physiochemical, structural, and immunological properties that can successfully trigger humoral and cellular immune responses against *K. pneumoniae*. Moreover, this vaccine can easily be over-expressed in *E. coli* strain K12. The promising computational results obtained notwithstanding, experimental validation of the designed multi-epitope vaccine is required to further verify the effectiveness of this potential vaccine candidate.

## Figures and Tables

**Figure 1 vaccines-07-00088-f001:**
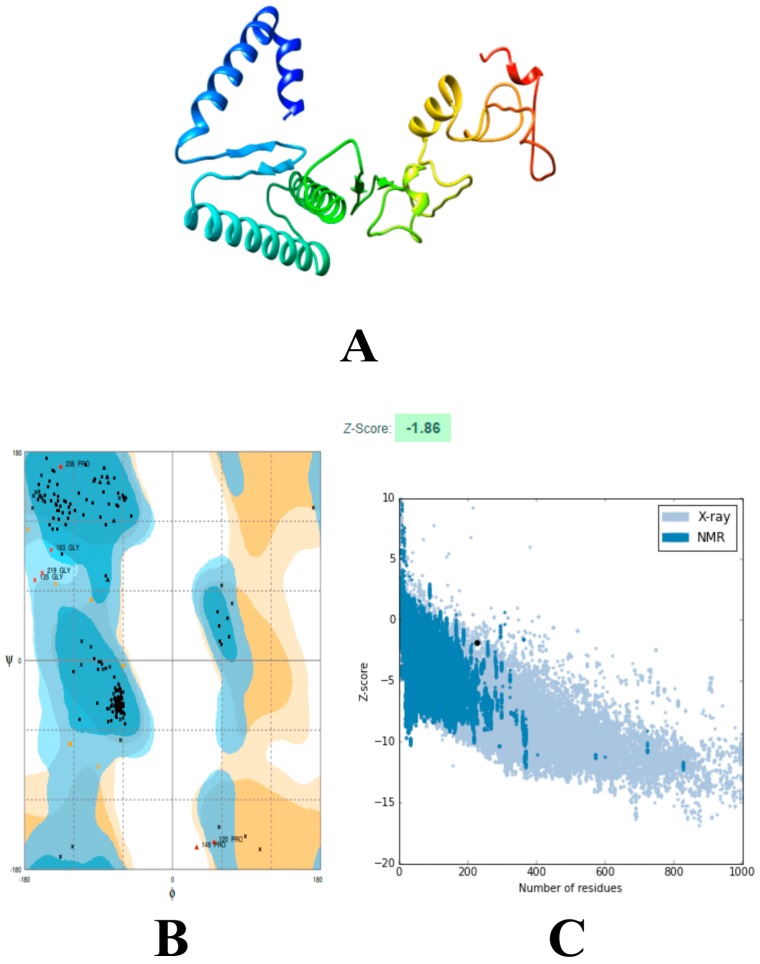
Structural analysis of designed vaccine. (**A**) The three-dimensional structure of vaccine obtained after molecular refinements; (**B**) Ramachandran plot analysis of the protein structure after molecular refinements. Analysis of the Ramachandran plot revealed that 94.7% of the residues of the vaccine are present in the favored region while 2.6% of the residues are present within the allowed and outlier regions, respectively; (**C**) ProSA-web evaluation of the vaccine structure. ProSA-web results indicated a Z-score of ‒1.86. The structure lies close to native X-ray resolved structures in the PDB (Protein Data Bank), which shows high-quality.

**Figure 2 vaccines-07-00088-f002:**
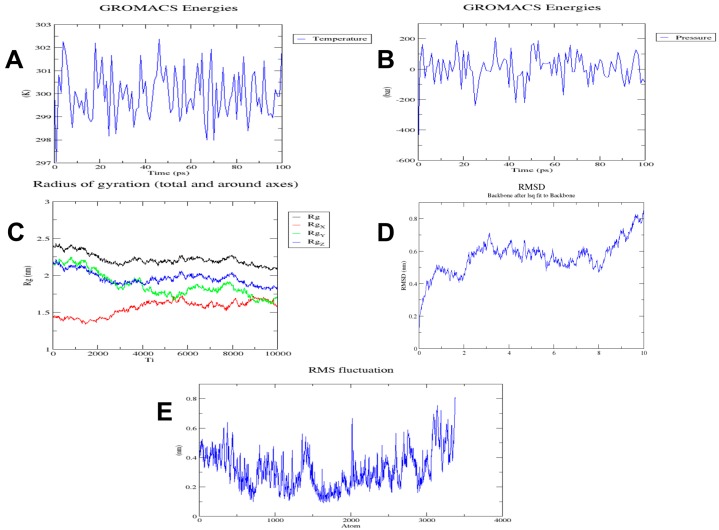
Graphs generated during different stages of MD simulations. (**A**) Temperature variations during simulation. System temperature reached 300 K and showed minimum fluctuations afterward during 100 ps; (**B**) pressure variations during simulation. Pressure plot shows that the average pressure is −2.44361 bar during 100 ps; (**C**) radius of gyration plot. Analysis of radius of gyration vaccine construct is stable in its compact form during the simulation time; (**D**) RMSD plot of backbone. RMSD graph shows that RMSD of protein backbone reaches ~0.6 nm and is maintained mostly, which represents minimum structural deviations of vaccine construct; (**E**) RMSF (Root Mean Square Fluctuation) plot. RMSF plot of side chains shows the regions with high flexibility present in peaks.

**Figure 3 vaccines-07-00088-f003:**
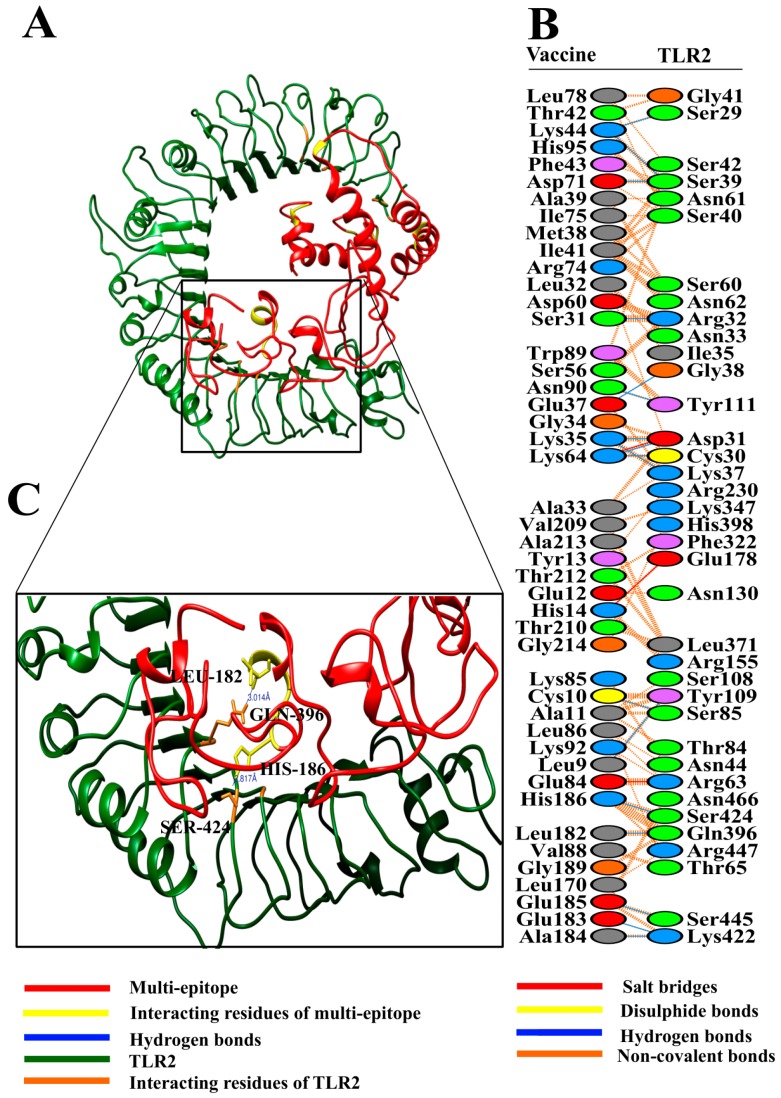
(**A**) Figure obtained after molecular docking showing vaccine construct-TLR2 docked complex. Vaccine construct is shown in red color while TLR2 is shown in green color; (**B**) interacting residues between docked vaccine (chain A) and TLR2 (chain B); (**C**) few prominent hydrogen bonds within vaccine-TLR2 complex are focused.

**Figure 4 vaccines-07-00088-f004:**
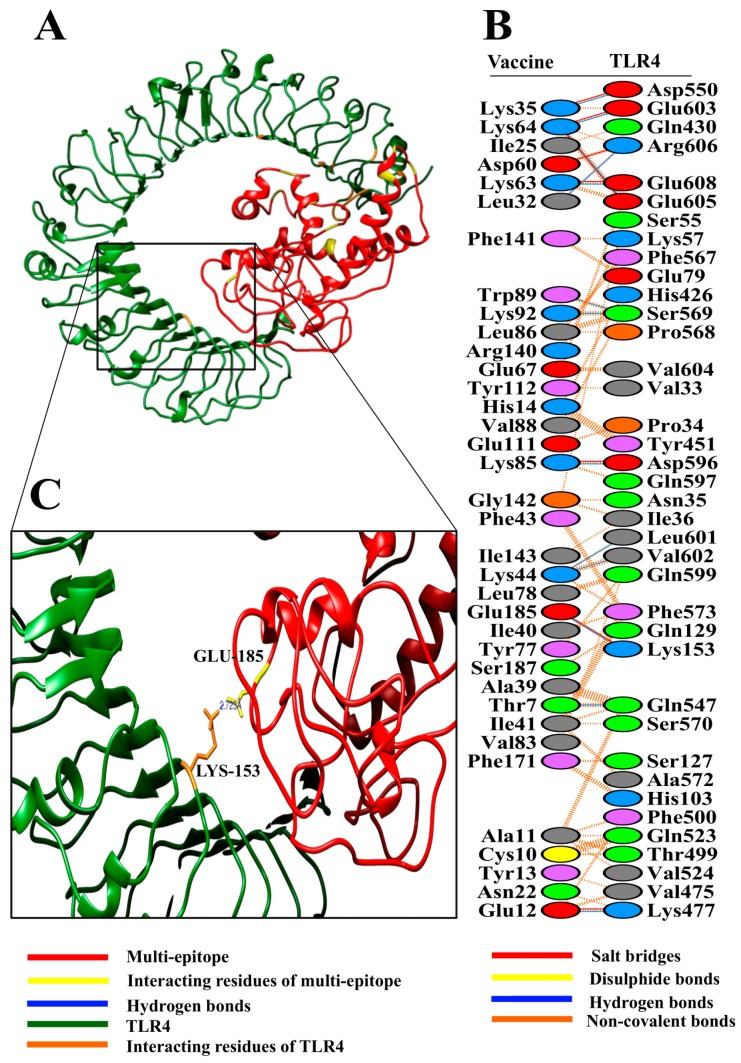
(**A**) Figure obtained after molecular docking showing vaccine construct-TLR4 docked complex. Vaccine construct is shown in red color while TLR4 is shown in green color; (**B**) interacting residues between docked vaccine (chain A) and TLR4 (chain B); (**C**) few prominent hydrogen bonds within vaccine-TLR4 complex are focused.

**Table 1 vaccines-07-00088-t001:** Table showing statistics of top vaccine-TLR2 docked cluster. Statistics of top ranked vaccine-TLR2 cluster are shown. Lower HADDOCK score signifies strong interaction between proteins.

Parameters	Value
HADDOCK score	−237.1 +/−3.3
Cluster size	20
RMSD from the overall lowest-energy structure	0.3 +/−0.2
Van der Waals energy	−156.8 +/−1.9
Electrostatic energy	−435.7 +/−19.0
Desolvation energy	6.8 +/−6.1
Restraints violation energy	0.0 +/−0.00
Buried Surface Area	4411.6 +/−24.1
Z-Score	0

**Table 2 vaccines-07-00088-t002:** Table showing statistics of top vaccine-TLR4 docked cluster. Statistics of top ranked vaccine-TLR4 cluster are shown. Lower HADDOCK score signifies strong interaction between proteins.

Parameters	Value
HADDOCK score	−235.6 +/−3.7
Cluster size	20
RMSD from the overall lowest-energy structure	0.3 +/−0.2
Van der Waals energy	−121.2 +/−3.5
Electrostatic energy	−517.3 +/−6.3
Desolvation energy	−11.0 +/−4.1
Restraints violation energy	0.0 +/−0.00
Buried Surface Area	4009.3 +/−30.5
Z-Score	0

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
