# Peer review of "Immunoinformatics-Aided Design and Evaluation of a Potential Multi-Epitope Vaccine against Klebsiella Pneumoniae"

_vaccines, 2019, doi:10.3390/vaccines7030088_

Round 1
Reviewer 1 Report
This manuscript by Dar et al. describes the conceptualization of a vaccine designed to prevent Klebsiella pneumoniae infections. The team take an immunoinformatics / reverse vaccinology approach to designing their vaccine.
There are many positives about the work. Overall it is well-conceived, the approach is thoughtful and comprehensive, and the manuscript is generally well-written and clearly presented.
The main weakness of the work is that the vaccine in question remains entirely conceptual without any physical interrogation or validation. While there is a track record for reverse vaccinology contributing to successful vaccine development (e.g. MenB) those efforts typically involved extensive, early testing of candidate molecules. Predictions notwithstanding it is unclear whether the candidate antigens in question are actually immunogenic, whether they can really be synthesized, whether they are indeed related to virulence, or even if they are expressed by Klebsiella during infection.
Even if immunization experiments are not feasible at this stage, the team could nonetheless validate their choice of target antigens. The simplest way of doing this would be to show that sera from human patients or mice experimentally infected with Klebsiella binds these antigens. if possible the team should do this.
It is also unclear in what way this work breaks new ground given that most of the software packages used are commercially available and the sequences in question are from public databases. Moreover, similar approaches have been developed for use with other organisms and also for klebsiella species (PMID:28570984). This reference and analogous work with other Gram negative pathogens should be cited and explored in the discussion.
Author Response
Point 1: The main weakness of the work is that the vaccine in question remains entirely conceptual without any physical interrogation or validation. While there is a track record for reverse vaccinology contributing to successful vaccine development (e.g. MenB) those efforts typically involved extensive, early testing of candidate molecules. Predictions notwithstanding it is unclear whether the candidate antigens in question are actually immunogenic, whether they can really be synthesized, whether they are indeed related to virulence, or even if they are expressed by Klebsiella during infection.
Response point 1: We thank the reviewer for the review and valuable feedback. We agree with the reviewer’s suggestion that extensive, early testing of candidate antigens is required to establish their immunogenicity. That is why we conducted a detailed literature search to find previous studies that provide information about the virulence, essentiality and antigenic nature of putative proteins identified in this study. Total two out of four proteins prioritized in this study outer membrane protein A and peptidoglycan-associated lipoprotein are well-known and characterized antigens of Klebsiella pneumoniae. The remaining two proteins Outer membrane porin PhoE and copper/silver efflux protein CusC also perform crucial cellular functions and are potential antigens based on our analysis. The relevant text to address point 1 of reviewer is provided in detail, in the revised manuscript in the discussion section from line 400-418.
“Two of the prioritized antigens in this study Peptidoglycan-associated lipoprotein (Pal) and Outer membrane protein A (OmpA) are recognized as dominant OMPs released by gram-negative bacteria during sepsis [70]. Kurupati et al. identified OmpA and other proteins using immunoproteome analysis to identify potential antigens of K. pneumoniae [71]. In another study, Kurupati et al. vaccinated mice using plasmid DNA containing ompA or ompK36 of K. pneumoniae which generated humoral as well as Th1 cell-mediated immune responses [72]. Similarly, Pal protein of K. pneumoniae is involved in protection against serum killing and neutrophil-mediated phagocytosis [73]. Additionally, it plays an important role in preserving the integrity of cell envelope and imperative for K. pneumoniae virulence in vivo. That study suggested that K. pneumoniae Pal can be used as attenuated vaccine in future.
Outer membrane porin-encoding gene PhoE is known as a housekeeping gene used for the molecular typing of K. pneumoniae [74]. According to information in the UniProt, PhoE expression is induced by phosphate starvation and it is involved in the uptake of inorganic phosphate, phosphorylated compounds, and other negatively charged solutes. Considering these crucial functions, PhoE protein also represents a good therapeutic target to tackle K. pneumoniae infection. The fourth proposed candidate copper/silver efflux system protein CusC is a part of cation efflux system that exports silver or copper ions from the periplasm via an antiport mechanism [75]. This overcomes the challenge posed by therapeutic usage of silver and copper and may help K. pneumoniae survive under these stress conditions”.
Point 2: Even if immunization experiments are not feasible at this stage, the team could nonetheless validate their choice of target antigens. The simplest way of doing this would be to show that sera from human patients or mice experimentally infected with Klebsiella binds these antigens. if possible the team should do this.
Response point 2: We highly value the insightful comments of the reviewer. Indeed, the experimental validation of putatively identified antigen through immunization experiments is recommended to verify the computer-aided analyses conducted in this study. However, unfortunately, we currently do not have the required funding and laboratory resources to undertake immunization experiments at this stage. We plan to perform these wet laboratory tests in the future subject to the availability of funds and lab equipment etc.
Point 3: It is also unclear in what way this work breaks new ground given that most of the software packages used are commercially available and the sequences in question are from public databases. Moreover, similar approaches have been developed for use with other organisms and also for klebsiella species (PMID:28570984). This reference and analogous work with other Gram-negative pathogens should be cited and explored in the discussion.
Response point 3: We appreciate the reviewer’s comments and tried to improve the manuscript quality after careful revision. The reviewer has questioned about the significance of our work. In this study, we have analyzed total 222 complete genomes of K. pneumoniae to identify four highly conserved antigens. Furthermore, we explored the candidate antigens using immunoinformatics approaches to construct a broad-spectrum multi-epitope vaccine against K. pneumoniae. To address reviewer’s concerns regarding the citation of similar studies targeting other Klebsiella species and additional gram-negative pathogens, we included the relevant points in the discussion portion of the manuscript, from line 424-428. The relevant text to address point 3 of reviewer is provided in detail, in the revised manuscript in the discussion section from line 419-428.
“In this study, we have used an in-house pipeline “VacSol” to identify potential antigens in K. pneumoniae and the candidates were verified through existing experimental data where applicable. Based on our experience and existing knowledge, we believe that these predicted proteins are good vaccine candidates. They are highly conserved proteins present in maximum if not all K. pneumoniae strains, are localized in the outer membrane and thus accessible to the host immune system, are known to be associated with virulence and crucial for the survival of bacteria. Similar approach based on reverse vaccinology has also been applied for the identification of vaccine targets in K. oxytoca, another closely related species to K. pneumoniae [76]. Furthermore, this approach has been adopted to target other gram-negative pathogens such as Acinetobacter baumannii [28,77], Pseudomonas aeruginosa [78], Neisseria gonorrhoeae [79], etc.”
Reviewer 2 Report
The authors' study attempts to identify potential proteins to serve as vaccine candidates against K. pneumoniae. Utilizing an in silico approach that included genomic and proteomic databases and molecular modeling programs, the authors have identified potential epitopes of interest to construct a vaccine. The study employs current technologies to begin addressing a significant medical challenge in treating/preventing severe infections.
Overall, the manuscript should be of interest to readers in multiple fields. I have no major concerns with the manuscript as submitted.
Minor concerns are as follows:
Line 73 - "retrieval" is misspelled
Line 84 - extra spaces following the word "Next,"
Line 104 - no space in MHCII
Line 164 - extra spaces following the word "compatible"
Line 217 - should the word "filer" actually be "filter"?
Line 257 - The (Chawley et al., 2014) reference does not follow the formatting for all the rest of the citations and is also missing from the References list at the end of the manuscript
Line 258-9 - Escherichia coli needs to be italicized
Line 334 - The table is describing the statistics for docking with TLR4, but the description says TLR2
Line 390 - Nonetheless should not be capitalized
Author Response
Reviewer comment: Overall, the manuscript should be of interest to readers in multiple fields. I have no major concerns with the manuscript as submitted. Minor concerns are as follows:
Line 73 - "retrieval" is misspelled
Line 84 - extra spaces following the word "Next,"
Line 104 - no space in MHCII
Line 164 - extra spaces following the word "compatible"
Line 217 - should the word "filer" actually be "filter"?
Line 257 - The (Chawley et al., 2014) reference does not follow the formatting for all the rest of the citations and is also missing from the References list at the end of the manuscript
Line 258-9 - Escherichia coli needs to be italicized
Line 334- The table is describing the statistics for docking with TLR4, but the description says TLR2
Line 390 - Nonetheless should not be capitalized
Response to the reviewer: We are pleased to know that you have found our research work interesting. In order to address your minor concerns, we have revised the manuscript and implemented the following steps:
The spelling of “retrieval” has been corrected in line 72 of the revised manuscript. Extra spaces after the word “Next,” have been removed in line 82-83. A space between “MHC” and “II” has been added in line 103. Extra space after the word “compatible” in line 163 has been removed. The spelling of “filter” has been corrected in line 215. The (Chawley et al., 2014) reference in line 255 has been replaced by another appropriate reference. The in-text and bibliographic citation of this new reference has been updated in the manuscript. Escherichia coli has been italicized in lines 256-257. Table 2 caption in line 331 has been corrected to indicate “TLR4” statistics. The capital letter in “Nonetheless” has been removed in line 387.Round 2
Reviewer 1 Report
The manuscript remains a conceptual exercise without any validation of the vaccine targets or other physical experiments. The additional references are helpful but the novelty and impact of this work are unclear.